# Comparative Analysis of Developmental Transcriptome Maps of *Arabidopsis thaliana* and *Solanum lycopersicum*

**DOI:** 10.3390/genes10010050

**Published:** 2019-01-15

**Authors:** Aleksey A. Penin, Anna V. Klepikova, Artem S. Kasianov, Evgeny S. Gerasimov, Maria D. Logacheva

**Affiliations:** 1Institute for Information Transmission Problems of the Russian Academy of Sciences, Bolshoy Karetny per. 19, build. 1, Moscow 127051, Russia; a.klepikova@skoltech.ru (A.V.K.); gerasimov_e@mail.bio.msu.ru (E.S.G.); m.logacheva@skoltech.ru (M.D.L.); 2Lomonosov Moscow State University, Leninskye Gory, Moscow 119992, Russia; 3Skolkovo Institute of Science and Technology, Center for Data-Intensive Biology and Biomedicine, Nobelya Ulitsa 3, Moscow 121205, Russia; 4Vavilov Institute of General Genetics, Russian Academy of Sciences, Gubkina 3, Moscow 119991, Russia; artem.kasianov@vigg.ru

**Keywords:** tomato, *Solanum lycopersicum*, RNA-seq, transcriptome atlas

## Abstract

The knowledge of gene functions in model organisms is the starting point for the analysis of gene function in non-model species, including economically important ones. Usually, the assignment of gene functions is based on sequence similarity. In plants, due to a highly intricate gene landscape, this approach has some limitations. It is often impossible to directly match gene sets from one plant species to another species based only on their sequences. Thus, it is necessary to use additional information to identify functionally similar genes. Expression patterns have great potential to serve as a source of such information. An important prerequisite for the comparative analysis of transcriptomes is the existence of high-resolution expression maps consisting of comparable samples. Here, we present a transcriptome atlas of tomato (*Solanum lycopersicum*) consisting of 30 samples of different organs and developmental stages. The samples were selected in a way that allowed for side-by-side comparison with the *Arabidopsis thaliana* transcriptome map. Newly obtained data are integrated in the TraVA database and are available online, together with tools for their analysis. In this paper, we demonstrate the potential of comparing transcriptome maps for inferring shifts in the expression of paralogous genes.

## 1. Introduction

In plant science, the overwhelming majority of experiments aimed at the identification of gene functions are carried out primarily in the model species, *Arabidopsis thaliana*. The knowledge of gene function is transferred from model species to non-model species (e.g., agriculturally important plants) based on the similarity of gene sequences and phylogenetic analysis. The assumption that underlies such transfer is that orthologous genes have similar functions. However, it is well-known that this is not true in many cases [1,2]). Additionally, the accuracy of orthology inference is not high, especially in plant genomes which are shaped by multiple whole-genome and segment duplications and subsequent gene loss. Thus, it is highly desirable to use additional information that would help in developing hypotheses on gene functions. Gene expression data can serve as a source of such information. The similarity of gene-expression profiles of homologous genes indicates the conservation of functions, while a drastic difference in expression profiles suggests functional divergence. This approach requires high-resolution data on gene-expression profiles that can be compared across species. Previously, we had developed a transcriptome map of the model plant *Arabidopsis thaliana* [3]; here, we report the transcriptome map of *Solanum lycopersicum* (tomato). Tomatoes are a representative of a large clade of eudicots—asterids, while *Arabidopsis* belongs to another large clade—rosids. Tomatoes are an important agricultural plant that are cultivated worldwide for its fruit; the gross production of tomato is >170 million tons [4]. Despite this, there is still great demand for new cultivars [5]. This requires an object-specific knowledge of gene functions. 

The (meta) analysis of high-throughput gene expression data has great potential to improve functional annotations (e.g., [6,7,8]). In December 2018, the National Center for Biotechnology Information (NCBI) database listed 213 BioProjects containing RNA-seq data for tomato; these projects included 3986 sequence read archive (SRA) accessions. The majority of these data can be divided into three categories. The first category represents fruit development: the main part of which is available via the TomExpress database (http://tomexpress.toulouse.inra.fr/ [9]), and the SolGenomics project (http://tea.solgenomics.net [10,11,12]), which provides an attractive and useful overview of gene expression in the tomato fruit; the second category represents studies on stress response; and the third includes studies comparing gene expression in wild-type plants and in mutants or genetically modified plants. Publicly available NCBI RNA-seq datasets that can be regarded as developmental transcriptome maps are represented by two BioProjects—PRJDB5790 and PRJNA307656 [13]. Unfortunately, both of them have an important shortcoming—they were conducted in one biological replicate. This hampers any statistical analysis of gene expression and splicing. The motivation for these studies was not genome-wide expression analysis. For example, in a study by Cárdenas et al. [13], the transcriptome map was used to study the function (including detailed expression profile and co-expression with other genes) of the gene GAME9. A global analysis of the transcriptome data was not performed. Taking into account this gap in the transcriptomic data of tomato, its taxonomic position, and agricultural importance, we have collected and sequenced tomato samples and created a tomato transcriptome atlas containing data on gene-expression profiles in 30 samples that represent different organs and developmental stages (Appendix A). The results are integrated in the TraVA (Transcriptome Variation Analysis) database (http://travadb.org/browse/Species=Tomato/). Here, we present the results of the analysis of these data, focusing on comparison with the *Arabidopsis* transcriptome map.

## 2. Materials and Methods

### 2.1. Sample Collection

Each sample is a pool collected from at least 10 plants in order to decrease the variance caused by inter-individual differences in gene expression. Samples were collected in two biological replicates. Plants were grown in a climate chamber (POL-EKO Aparatura, Vladislavia, Poland) under a 16 h light/8 h dark cycle at 22 °C and 50–60% relative humidity.

### 2.2. RNA Extraction

RNA was extracted using the RNeasy mini kit (Qiagen, Venlo, The Netherlands) and following the manufacturer’s protocol. RNA quality was controlled using capillary electrophoresis on a Bioanalyzer 2100 (Agilent, Santa Clara, CA, USA).

### 2.3. Library Preparation and Sequencing

PolyA mRNA was extracted using TruSeq RNA Sample Prep Kits v2 (Illumina, San Diego, CA USA) in 0.4 of the recommended volume, due to the small amounts of RNA in some samples. Illumina cDNA libraries were constructed with the NEBNext Ultra II RNA Library Prep Kit for Illumina (New England BioLabs, Ipswich, MA, USA) following the manufacturer’s protocol in 0.5 of the recommended volume. Sequencing of the cDNA libraries was performed using the HiSeq4000 (Illumina) instrument (4 lanes, 60 bp single read run). For several libraries, the resulting the total number of reads sequencing on HiSeq4000 was less than 20 million, and they were sequenced once more on NextSeq500 (Illumina) with a 75 bp read length.

### 2.4. Mapping

Raw reads from the SRA (Appendix A) were downloaded using the recommended utility fastq-dump (from NCBI SRA-toolkit version 2.8.0) with the “--split-files” option.

Reads for publicly available datasets from SRA and reads for 30 samples obtained in the frame of this study were quality- and adapter-trimmed using Trimmomatic [14] version 0.36. For single-read data, Trimmomatic was run in single-read mode, and for paired-end—in paired-end mode. Other Trimmomatic options (identical for single-read and paired-end data) were set with the following command line: “ILLUMINACLIP:common.adators.file:2:30:10 LEADING:20 TRAILING:20 SLIDINGWINDOW:4:15 MINLEN:30”.

Genome assembly for *S. lycopersicum* was taken from the Ensembl database: assembly and corresponding annotation version SL2.50. Trimmed reads were mapped on the genome assembly using Spliced Transcripts Alignment to a Reference (STAR) [15] version 2.4.2a in the “GeneCounts” mode, and with the provided annotation file to obtain read counts for genes. In this mode, STAR v. 2.4.2 is claimed to count only uniquely mapped reads per gene. Other mapping parameters were adjusted by the following options: “--outFilterMismatchNmax 3--outSJfilterCountUniqueMin 3 1 1 1—outSJfilterCountTotalMin 3 1 1 1--alignIntronMin 6”.

For the FRUIT dataset, see the results of mapping in Appendix A.

### 2.5. Splicing Analysis

Splicing analysis was based on the splice junctions’ output from STAR. Two filter criteria were applied to the collection of all discovered splice junctions for each dataset in the study. The splice junction passes filter 1 if it exists in two or more sequencing runs (e.g., any two fastq files). The splice junction passes filter 2 if it exists in two replicates of the sample (two fastq files from the same sample). For samples in the public DEVELOPMENT dataset (Appendix A), there were no replicates provided; thus, filter 2 was not used for this dataset. Filtering and counting were performed using custom Python script. The filtering procedure is crucial to remove possible artefacts, especially on the read-mapping stage of data processing. On the other hand, filtration that is too strict can remove rare splicing events from analysis (for example, in our development dataset 1730 (1,3%), introns annotated in SL2.50 did not pass even filter 1). The splice junctions are listed in Appendix A.

### 2.6. Expression Analysis

In order to quantify gene expression levels, we calculated total gene read (TGR) values. To avoid library size bias, TGR values were normalized between samples using size factors, as described by Anders and Huber [16]. A gene was considered as expressed if, in each biological replicate, the gene has a normalized TGR value of 5 or higher (weak threshold) or 16 or higher (strong threshold, [17]). For the completeness of the discovery of expressed genes, three publicly available datasets were used (Appendix A) with the same thresholds. Differentially expressed genes were detected using the R package “DESeq2” [18] with the following thresholds: a false discovery rate (FDR) <0.05 and a fold change >= 2. The differential expression (DE) score was defined as the number of pairwise comparisons in which a given gene was differentially expressed [3].

### 2.7. Detection of Stably Expressed Genes

For the assessment of expression stability, only genes expressed in all samples under a weak threshold were considered. Using normalized TGR values, the mean and standard deviation of expression were calculated for each gene expressed in all samples. The coefficient of variation (CV) was calculated as the standard deviation divided by the mean. Genes with CV less than 0.3 were considered stably expressed across all samples.

### 2.8. Gene Ontology Enrichment Analysis

Overrepresented Gene Ontology (GO) categories in gene lists in comparison with all genes of *S. lycopersicum* were found using the PANTHER Classification System Version 13.1 [19,20] statistical overrepresentation test with default settings (including FDR <0.05) and fold enrichment >= 2. PANTHER Pathways and the PANTHER protein class were also checked for overrepresentation.

### 2.9. Shannon Entropy

Shannon entropy (H) values were used for expression pattern width assessment, and were calculated for genes expressed in at least one sample under a weak threshold, as done by Schug et al. [21]. To avoid bias due to overrepresentation of certain parts and organs, the samples were grouped using hierarchical clustering: samples with a distance (1 − Pearson *r^2^*) <0.3 were grouped (the sample combination is described in Appendix A), and the gene expression levels were averaged.

### 2.10. Pseudo-Euclidean Distance

For cross-species TGR normalization, 7460 pairs of orthologous genes were taken. 79 samples in two biological replicates for 7460 *Arabidopsis* genes and 30 samples in two biological replicates for 7460 *Solanum* genes were combined in a single dataset, with 109 samples for 7460 genes. The TGR of this dataset was normalized by size factor as in “DESeq” [16], and the size factor values for each sample were stored and used for the normalization of expression of all genes in the *Arabidopsis* and *Solanum* datasets separately.

Only genes that were expressed in at least one sample were considered for analysis in both *Arabidopsis* and *Solanum*. For all genes, the normalized TGR was incremented by 1 to obtain non-zero values of median expression for tissue-specific genes; for each gene, the normalized TGR in all samples were divided by the median, which provides relative expression profiles.

Although the *Solanum* transcriptome map was collected in agreement with the *Arabidopsis* atlas, several samples were hard to match directly; such samples were grouped. A list of sample combinations is provided in Appendix A.

For each pair of *Arabidopsis* and *Solanum* genes, the pseudo-Euclidean distance was calculated in 100 repeats as follows:For each sample of *Arabidopsis* and *Solanum*, one of the biological replicates was randomly taken.For each pair of samples, the residuals of median-normalized TGR were calculated. In the case of a group of samples, the residuals were counted for all possible pairs of *Arabidopsis* and *Solanum* samples, and a minimum value of residuals was chosen.All residual values were summed, and a squared root of the sum was calculated to obtain the pseudo-Euclidean distance.

Then, 100 replicates of the pseudo-Euclidean distance were averaged and used as the expression distance measure for the pair of genes.

### 2.11. Orthology Assessment

For the detection of orthogroups, the OrthoFinder version 2.2.6 [22] software with default parameters was used. Proteins from the longest isoforms of the TAIR10 version of the *A. thaliana* annotation and the SL2.50 version of the *S. lycopersicum* annotation were used as OrthoFinder input.

### 2.12. Data Availability

The raw data of the tomato transcriptome map have been deposited into the NCBI Sequence Read Archive (project ID: PRJNA507622).

## 3. Results and Discussion

### 3.1. Sampling and Primary Analysis

The choice of samples for the tomato transcriptome map was based on clustering of *A. thaliana* transcriptome data from Klepikova et al. [3]. We selected the *Arabidopsis* samples that had the most dissimilar expression profiles based on the clustering tree of samples, and collected tomato samples that corresponded to these *Arabidopsis* samples (for example, anthers and senescent leaves). Assuming that expression profiles in homologous organs and/or corresponding developmental stages are similar in *Arabidopsis* and tomato, this approach would result in a set of tomato samples representing the maximum diversity of expression profiles. 

The samples were sequenced with at least 20 million sequence reads were generated for each sample and read length of 75 and 60 bp (see Materials and Methods). Initial quality analysis showed a high congruence of the biological replicates: Pearson *r^2^* correlation values for all replicates were between 0.79 and 1.0, with a mean value of 0.96 (median 0.98) (Appendix A), and a clustering tree of the replicates also indicated consistency of the data (Appendix A). A hierarchical clustering tree of the samples reflected an organ- and age-specific structure (Figure 1). Most samples which are not replicates have highly divergent expression profiles (1-*r^2^* >0.3). This shows that the initial assumption was true and that our map indeed represents samples which are the most diverse in terms of expression profiles.

Annotation SL2.50 of the *S. lycopersicum* genome contains 33,810 coding genes. We used two thresholds to define genes as expressed in a certain sample: five normalized read counts in each of two replicates of the sample (weak threshold), and 16 normalized read counts for the strong threshold (as defined by Su et al. [17]). Using the weak threshold, 26,283 (78%) of genes were expressed in at least one sample (24,792 (73%) using strong threshold, Appendix A). In all samples with weak and strong thresholds, 13,517 (40%) and 11,669 (35%) genes were expressed, respectively (Appendix A). The lowest number of expressed genes (17,208 (51%) and 15,348 (45%) for weak and strong thresholds, respectively) was observed in the Sol.FL.r sample (red pulp), while the greatest number (20,805, 62% and 18,564, 55%) was observed in the Sol.SD.y sample (young seeds) (Appendix A).

The splicing analysis demonstrates that the current annotation of the tomato genome lacks many splice sites. Our dataset reveals a high number of new splice sites. In contrast, only 10% of 123,617 previously known splice sites are not found in our data. Regarding new splice sites, even at the most stringent threshold, the number of new sites is twice as much as the number of annotated sites. The results of splicing are summarized in Table 1.

To assess the completeness of the transcriptome map in terms of the representation of expressed genes, we used three publicly available datasets that represent different biological processes and organs. The complete list of samples is presented in Appendix A. The first dataset—DEVELOPMENT—includes 19 samples (floral bud, leaf, petal, root, and different parts of the fruit at five stages of fruit maturity) in one replicate with a sequencing depth of 14–26 million reads [13]. The second dataset—STRESS—includes two sets of samples from biotic stress (*Cladosporium fulvum* infection-treated and control plants [23] and PRJNA419151). Each set is a time series collected in three replicates, and the sequencing depth ranges between 9.7–31 million reads. The third dataset—FRUIT—is a detailed expression atlas of the developmental dynamics of the tomato fruit [12]. It includes 49 samples in three replicates and 84 samples in four replicates. The sequencing depth is moderate, ranging from 3.6 to 25 million reads. Out of 483 samples, 183 have more than 10 million, and 367 have more than 7 million reads. The total number of genes expressed in the samples from these three datasets is 27,562 (under the threshold 5+5 reads); in our transcriptome map, we registered the expression of 26,283 genes (i.e., >95%). The same pattern is retained under a stronger threshold—16+16 reads: out of 25,908 genes expressed in these three datasets, and 24,792 genes are observed in our map. The expression of ~1000 genes is registered only in our dataset (Figure 2a); we assume that this is because several samples are unique in our map, e.g., meristems.

Next, we assessed the number of samples in which each gene was expressed (Appendix A). Most of the protein-coding genes tended to be expressed in all or almost all samples (16,326, 48% (14,378, 43%) genes were expressed in more than 25 samples), while some genes were expressed in a few samples (3365, 10% (3674, 11%) genes in 1–7 samples). We also investigated whether there was a correlation between the number of samples in which a gene was expressed and the expression level (Appendix A). The mean and median expression levels across all samples were found to be higher for more widely expressed genes (i.e., those expressed in more samples). For maximum and minimum expression levels, the most widely expressed genes also exhibited a greater expression level, but the trend was not as prominent for these genes.

Analysis of splice sites using additional publicly available datasets shows that even in our dataset, many low-frequency sites remain unidentified. In particular, the addition of the detailed transcriptome map of fruit development results in a high number of additional splice sites (Figure 2b and Table 1). However, given low coverage of the data, they may represent artefacts.

### 3.2. Comparison with Arabidopsis thaliana Transcriptome Map

We compared the global parameters of the tomato transcriptome map with those of the *A. thaliana* map and found that, despite the difference in number of samples, they were similar in these two species. In particular, the distribution of the number of expressed genes and Shannon entropy (H) are similar, with the only difference being that in tomato, the peak at low entropy values is almost not visible (Appendix A). The maximum entropy is 4.16. There are 12,641 genes with H >= 3.7; they are highly enriched in terms of being associated with basic cellular metabolism (Appendix A). At the lower end, there are 298 genes with H <= 0.15. They are enriched in categories such as peroxidase activity (GO:0004601) and peptidase activity (GO:0008233) in molecular function or response to stress (GO:0006950) in biological processes or the cell wall (GO:0005618) in cellular components (Appendix A). All other global parameters (see Appendix A), such as distribution of maximum and minimum expression levels, DE score, and Z-score, are also almost identical in tomato and *Arabidopsis*.

It is interesting to compare a set of genes that do not vary in expression between samples in *Arabidopsis* and in tomato. We considered only genes expressed in all samples; for each gene, a covariation was calculated. We found 123 genes with CV <0.20, 657 with CV <0.25, and 1527 with CV <0.30 (Appendix A). A set of genes with CV <0.2 was enriched by the categories related to transport, protein, nucleic acid localization, and kinases (Appendix A). Similar to *Arabidopsis*, the addition of publicly available RNA-seq data (sets DEVELOPMENT and STRESS) did not greatly decrease the number of stable genes (Appendix A). Unfortunately, the data from the fruit development atlas could not be used for the analysis of stable genes due to shallow sequencing depth that can lead to distortion in expression profiles (in particular, the underestimation of lowly expressed genes).

Analysis of GO enrichment of stable genes in *Arabidopsis* and tomato reveals similar categories: GO:0051169~nuclear transport, GO:0016192~vesicle-mediated transport, GO:0015031~protein transport, GO:0008104~protein localization, GO:0006886~intracellular protein transport, GO:0006497~protein amino acid lipidation, GO:0006403~RNA localization, GO:0006397~mRNA processing GO:0004386~helicase activity, GO:0042175~nuclear envelope-endoplasmic reticulum network, GO:0016023~cytoplasmic membrane-bounded vesicle, GO:0005794~Golgi apparatus, GO:0005654~nucleoplasm, and GO:0005635~nuclear envelope. 

The identification of stably expressed genes is important for further studies that utilize quantitative PCR (qPCR) for the measurement of gene expression levels. It is well-known that the genes traditionally used as a reference in qPCR experiments (glyceraldehyde 3-phosphate dehydrogenase (GAPDH), actin, tubulin, etc.) are indeed not stable across conditions and organs [24], and each species and each experimental system requires selection and validation of the optimal reference genes [25]. The set of stably expressed genes identified in our study could be used as a basis for such a selection in tomato. Notably, tomato orthologues of two genes that were identified as the most stable in *Arabidopsis* are also among the most stable (Appendix A).

### 3.3. Analysis of Expression Patterns of Duplicated Genes

The most prominent feature of plant genomes is that they undergo multiple whole-genome or segmental duplications. Gene copies resulting from the duplication usually diverge in functions (alternatively, one of the copies can be lost). In cases when one gene of a model object is an orthologue of two paralogous genes from a non-model object, it is usually difficult to identify which of the co-orthologues retains ancestral function because both of them have a similar level of sequence identity. Indeed, we found that the distribution of the identity values for interspecific pairs within ortho-triplets (1 *Arabidopsis* gene–2 tomato genes and vice versa) is almost identical with the distribution of the identities for orthopairs. Even the distributions of minimal and maximal identity values are not drastically different (Figure 3 and Table 2).

This means that in most cases of ortho-triplets, both paralogs from one species are equally similar to a single gene from the other species. In contrast, the similarity of expression profiles greatly differs for interspecific pairs—i.e., for most ortho-triplets, there is a pair with low expression distance (close to the distance typical for ortho-pairs) and a pair with high distance (see e.g., Figure 4 and Table 2).

In terms of function, this means that one of the co-orthologues in the ortho-triplet usually retains the ancestral function, while the other acquires a new function. Presumably, this occurs by the divergence of the regulatory elements of the paralogs after duplication. At the same time, sequence similarity at the level of protein-coding sequencing remains the same for both co-orthologues, and does not allow for conclusions on the function to be made.

### 3.4. Integration of the Solanum Transcriptome Map into the Database TraVA

Our expression data were integrated into the public database, Transcriptome Variation Analysis (TraVA, [3]). It allows for graphical representation of the expression profiles of single or multiple genes (delimited by spaces), both in absolute and relative to maximum values. A user can also choose a type of read count normalization. The results of differential expression analysis were also included in TraVA. A user can select a sample of interest from a list of samples and receive a colored visualization of fold changes for the analyzed gene, or genes between the selected sample and all other samples. The results of all types of analyses can be downloaded as Excel files.

## 4. Conclusions

The transcriptome map includes expression data for over 95% of the genes annotated in the tomato genome, allowing for the analysis of differential expression between organs and stages. Our previously developed *Arabidopsis* transcriptome map that has a similar sample structure allows one to conduct side-by-side comparison of gene expression profiles, and to reveal the cases of conservation and shift of function in co-orthologues.

## Figures and Tables

**Figure 1 genes-10-00050-f001:**
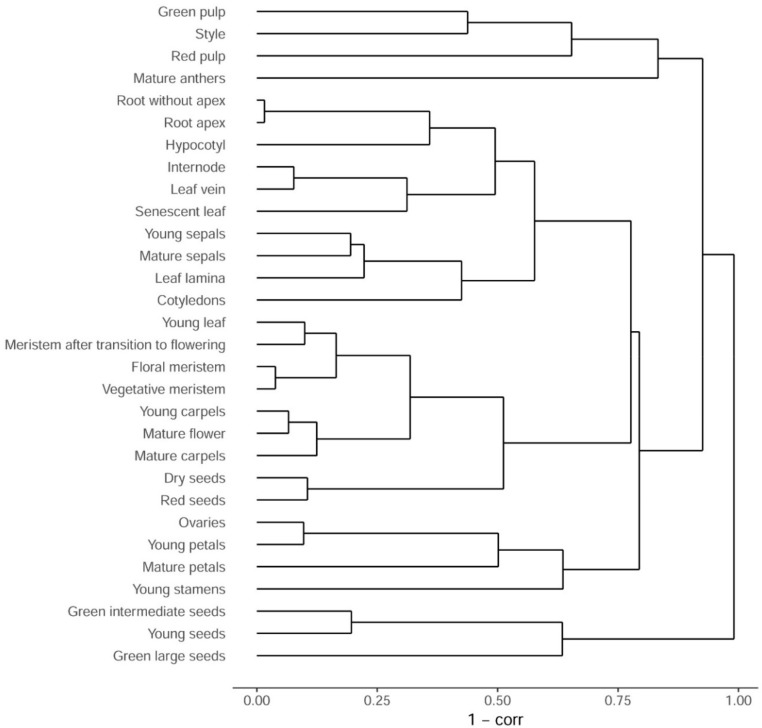
Hierarchical clustering of samples as represented by a clustering tree. Distance between samples is measured as 1 − Pearson *r*^2^ correlation coefficient.

**Figure 2 genes-10-00050-f002:**
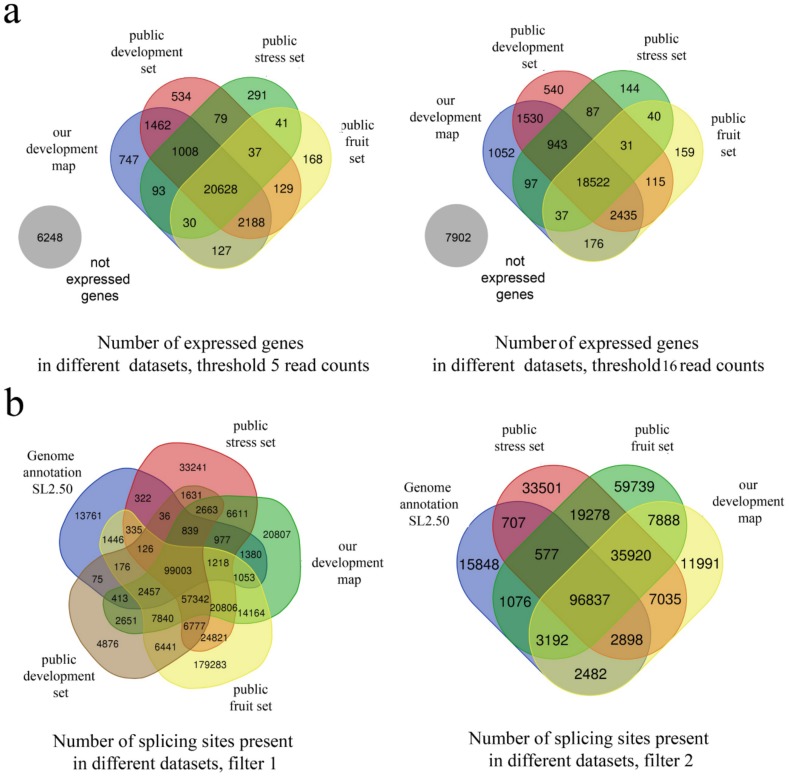
Estimation of the completeness of the expression map. (**a**) The number of expressed genes at different thresholds in three publicly available datasets and in our dataset; (**b**) the number of detected splice sites under different filters in three publicly available datasets and in our dataset.

**Figure 3 genes-10-00050-f003:**
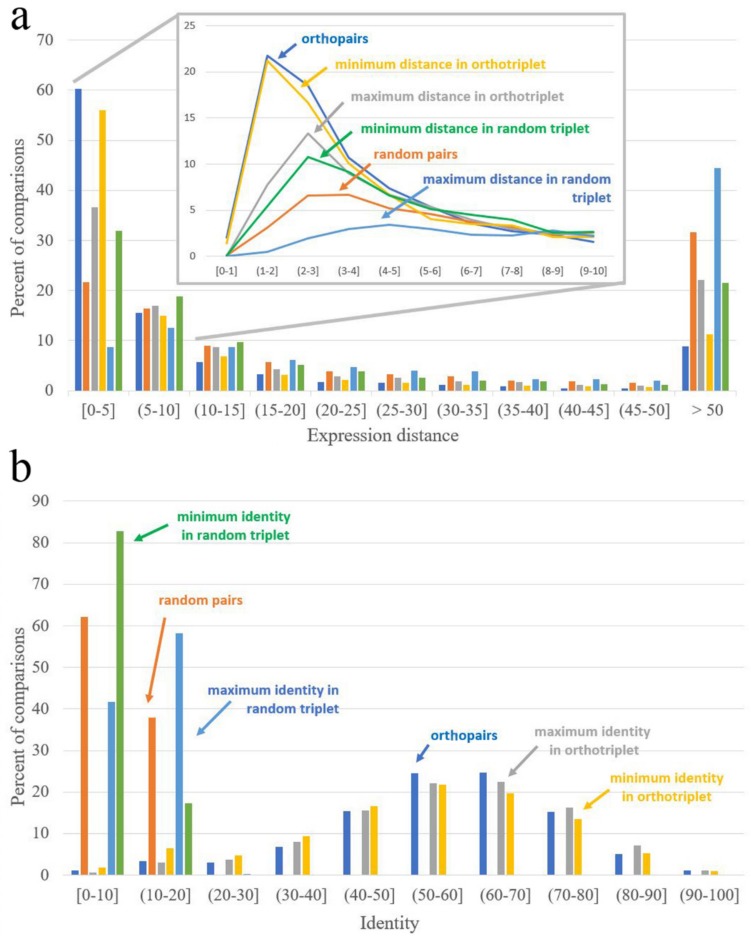
Distribution of identity and expression distance for orthopairs, random pairs, interspecific pairs from ortho-triplets, and interspecific pairs from random triplets. (**a**) The distribution of expression distance; (**b**) the distribution of identity.

**Figure 4 genes-10-00050-f004:**
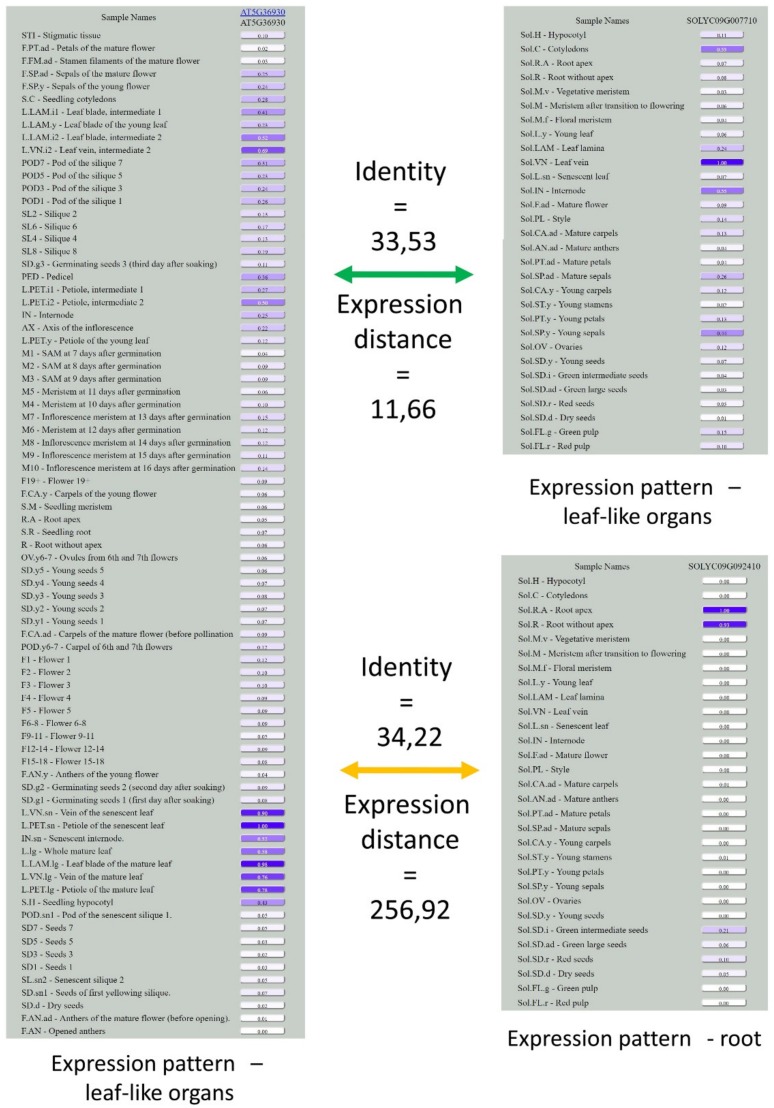
Expression patterns in ortho-triplet AT5G36930, Solyc09g092410, and Solyc09g007710. Identities are almost equal, while expression distances differ ~20-fold. Note that while Solyc09g007710, similar to AT5G36930, is expressed in leaves, Solyc09g092410 has a drastically different expression profile, with a maximum expression level in roots.

**Table 1 genes-10-00050-t001:** Analysis of splicing sites based on newly obtained data.

	Without Filtering	Filter 1 (Identification in Two Samples)	Filter 2 (Identification in Two Replicates)
Introns, total	375,650	240,224	168,243
Not annotated but found	266,580	132,884	62,834
Annotated but not found	14,547	16,277	18,208

**Table 2 genes-10-00050-t002:** Medians of distributions of identity and expression distance.

Expression Distance Distance = 0 Corresponds to Identical Expression Patterns		Identity Identity = 100 Corresponds to Identical Sequences	
Orthopairs	3.68	Orthopairs	58.48
Random pairs	17.42	Random pairs	8.40
Minimal distance in interspecific pairs from ortho-triplets	4.12	Maximal identity in interspecific pairs from ortho-triplets	58.78
Maximal distance in interspecific pairs from ortho-triplets	8.52	Minimal identity in interspecific pairs from ortho-triplets	55.01
Minimal distance in interspecific pairs from random triplets	9.76	Maximal identity in interspecific pairs from random triplets	10.96
Maximal distance in interspecific pairs from random triplets	37.64	Minimal identity in interspecific pairs from random triplets	6.18

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
