# Peer review of "Comparative Analysis of Developmental Transcriptome Maps of *Arabidopsis thaliana* and *Solanum lycopersicum"

_genes, 2019, doi:10.3390/genes10010050_

Reviewer 1 Report

Manuscript Genes-410128 entitled “Comparative analysis of development transcriptome maps of Arabidopsis thaliana and Solanum lycopersicum” by Penin et al constructed a transcriptome atlas of tomato and compared with that of Arabidopsis.

In general, this manuscript is sound and could be an important resource for the tomato research community. However, I think it needs professional English editing, because many parts were not clearly written and the use of verb tense was not always consistent throughout the manuscript. It was therefore, difficult to review this properly. I provide some suggestions that I found when looked through it.

1.      Line 27: was that 30 organs at different developmental stages (meaning more than 30 samples were used) or 30 samples altogether?

2.      Line 36, add a comma after “In plant science”.

3.      In general, I think the introduction was poorly written and difficult to follow. The author should briefly summarize with more details the data in Supplementary Table S1, which they used for the database construction. The publication sources should be cited in the main text.

4.      Line 53: please add a reference for the figure that was mentioned (170 million tons).

5.      Line 56, it would be better to add a specific date to the sentence, for example, “as of December 2018”, instead of “Currently”. And also, add a comma after "currently".

6.      Line 61: what does “datasets” refer to?

7.      Line 62: it is not clear to me why the authors mentioned only 2 BioProjects here, while in Supplementary Table 1, it has more than two?

8.      Line 63: change to “they were”

9.      Line 64: change to "this study"

10.   Line 69: change to “agricultural importance, we created”

11.   Line 83: what does “read length 75/60 bp” mean?

12.   Figure 2a : mis-spelled "number"

13.   Line 112: data presented in this paragraph would be easier to follow if accompanied with a summary table, a short version of Supplementary Table S1.

14.   Line 122: change to “and 367 had more than 7 million reads".

15.   Line 134 and 135: would be helpful to if the authors could provide percentage of expressed genes out of the total numbers.

16.   Line 150: define “H” in the first use, for example, use “Shannon entropy (H)” in line 148

17.   Line 239: the sentence needs more work.

18.   241: Other trimmers? The settings are still for Trimmomatic, what does “other trimmers” refer to here?

19.   The authors should cite the sources for the softwares they used, including Trimmomatic (line 239), STAR (line 245).

20.   Line 266: spell out "TGR" in the first use.

21.   Line 268: the authors have not define "DEV" previously.

22.   Are the cutoffs for weak threshold and strong threshold the same for the 3 datasets? Do 5 and 16 in line 268 also mean 5 or more, and 16 or more, respectively? If so, this paragraph can be revised to state that once for thee datasets.

23.   Should the conclusion section be before the Materials and Methods?

Author Response

Reviewer 1

Manuscript Genes-410128 entitled “Comparative analysis of development transcriptome maps of Arabidopsis thaliana and Solanum lycopersicum” by Penin et al constructed a transcriptome atlas of tomato and compared with that of Arabidopsis.

In general, this manuscript is sound and could be an important resource for the tomato research community. However, I think it needs professional English editing, because many parts were not clearly written and the use of verb tense was not always consistent throughout the manuscript. It was therefore, difficult to review this properly. I provide some suggestions that I found when looked through it.

We agree. The text is now edited by the specialists of the language editing service American journal experts.

https://secure.aje.com/download.php?action=certificate&key=00E8-EBFA-1DC6-3FDB-307E&_t=1546212364728

1.      Line 27: was that 30 organs at different developmental stages (meaning more than 30 samples were used) or 30 samples altogether?

30 samples altogether. The description of samples in now included in the Supplementary Table S1

2.      Line 36, add a comma after “In plant science”.

This is done

3.      In general, I think the introduction was poorly written and difficult to follow. The author should briefly summarize with more details the data in Supplementary Table S1, which they used for the database construction. The publication sources should be cited in the main text.

We revised the introduction and cited the publications reporting the data used as our control datasets (“development”, “stress” and “fruit”). However, in this manuscript we focus on our newly obtained data. Only those data were used for database construction; the previously published data were used only for the assessment of the completeness of the representation of expressed genes and splice sites.

4.      Line 53: please add a reference for the figure that was mentioned (170 million tons).

This is done.

December 2018”, instead of “Currently”. And also, add a comma after "currently".5.      Line 56, it would be better to add a specific date to the sentence, for example, “as of

This is done.

6.      Line 61: what does “datasets” refer to?

We meant publicly available RNA-seq datasets. This is clarified now.

7.      Line 62: it is not clear to me why the authors mentioned only 2 BioProjects here, while in Supplementary Table 1, it has more than two?

This is done.

The description of samples in now included in the Supplementary Table S2

8.      Line 63: change to “they were”

This is done.

9.      Line 64: change to "this study"

This is done.

10.   Line 69: change to “agricultural importance, we created”

This is done.

11.   Line 83: what does “read length 75/60 bp” mean?

The data were obtained using two Illumina instruments - Hiseq4000 and Nextseq500. Read length was 75 for Nextseq and 60 for Hiseq. We added the explanation in materials and methods.

12.   Figure 2a : mis-spelled "number"

This is done.

13.   Line 112: data presented in this paragraph would be easier to follow if accompanied with a summary table, a short version of Supplementary Table S1.

This is corrected.

The description of samples in now included in the Supplementary Table S2

14.   Line 122: change to “and 367 had more than 7 million reads".

This sentence is corrected.

15.   Line 134 and 135: would be helpful to if the authors could provide percentage of expressed genes out of the total numbers.

This is done.

16.   Line 150: define “H” in the first use, for example, use “Shannon entropy (H)” in line 148

This is done.

17.   Line 239: the sentence needs more work.

We rephased this sentence making it more clear.

18.   241: Other trimmers? The settings are still for Trimmomatic, what does “other trimmers” refer to here?

We meant “other trimming options”. This is corrected now.

19.   The authors should cite the sources for the softwares they used, including Trimmomatic (line 239), STAR (line 245).

This is done.

20.   Line 266: spell out "TGR" in the first use.

This is done.

21.   Line 268: the authors have not define "DEV" previously.

This is done.

22.   Are the cutoffs for weak threshold and strong threshold the same for the 3 datasets? Do 5 and 16 in line 268 also mean 5 or more, and 16 or more, respectively? If so, this paragraph can be revised to state that once for thee datasets.

This is done.

23.   Should the conclusion section be before the Materials and Methods?
This is done.

Reviewer 2 Report

Line 19. This might be just an opinion, so do not consider it as a criticism, when you refer to "objects" do you mean organisms? I like much better to call it organisms than objects.

Line 30. I have been using your TraVA website for Arabidopsis lately.  In this article, as well as in your website, you state that several genes can be queried at once but it is not clear how to introduce the gene ids. Could you please maybe clarify this?

Line 36. Replace "on" by "of".

Line 37. Replace "a" by "the".

Line 58. When the different publicly available data sets used in the study are presented, one of the Solgenomics data set is not mentioned. Later on in the results and discussion section of the article, the Solgenomics data set is not used either. Incorporating the information from the Solgenomics set of data would improve the analysis presented in your article. Could you introduce that?

Line 182. Analysis of duplicated genes section. Totally agree with the authors that analysis of the gene expression patterns is perhaps the best way to test gene function across species. However, synteny analysis can be also very informative as well as, comparing the gene structures of the genes and assessing if exon and introns structures and sizes are conserved. This kind of information would help to confirm your tissue-specific expression analysis of putative orthologs among species. Could you introduce some of this analysis? 

Author Response

Line 19. This might be just an opinion, so do not consider it as a criticism, when you refer to "objects" do you mean organisms? I like much better to call it organisms than objects.

 We agree, this is corrected.

Line 30. I have been using your TraVA website for Arabidopsis lately.  In this article, as well as in your website, you state that several genes can be queried at once but it is not clear how to introduce the gene ids. Could you please maybe clarify this?

Gene names should be divided by spaces. We included this information in the manuscript.

Line 36. Replace "on" by "of".

This is corrected.

Line 37. Replace "a" by "the".

This is corrected.

Line 58. When the different publicly available data sets used in the study are presented, one of the Solgenomics data set is not mentioned. Later on in the results and discussion section of the article, the Solgenomics data set is not used either. Incorporating the information from the Solgenomics set of data would improve the analysis presented in your article. Could you introduce that?

As soon as we focus on side-to-side comparison with Arabidopsis we do not incorporate publicly available datasets into our database. We use them for the assessment of the completeness of the representation of expressed genes and splice sites. In this analysis the Solgenomics data are used (FRUIT set).

Line 182. Analysis of duplicated genes section. Totally agree with the authors that analysis of the gene expression patterns is perhaps the best way to test gene function across species. However, synteny analysis can be also very informative as well as, comparing the gene structures of the genes and assessing if exon and introns structures and sizes are conserved. This kind of information would help to confirm your tissue-specific expression analysis of putative orthologs among species. Could you introduce some of this analysis? 

We agree that structural information can also be useful for inferring gene functions. It is however better suited for more closely related species, where alignments of large genome segments can be generated. Also, it is more dependent on the quality of genome annotation and assembly. This is obviously not the problem in case of Arabidopsis and tomato. But we think of our manuscript not only as of a report on these two species but also as of an example of approach based on the comparison of expression maps - and expression maps can be generated for organisms that lack high quality genome assemblies. Thus in the current manuscript we would prefer to focus on gene expression patterns.

Reviewer 3 Report

The manuscript “Comparative analysis of development transcriptome maps of Arabidopsis thaliana and “Solanum lycopersicum” by Penin et al. constructs a transcriptomic expression database for tomato. The results of the manuscript greatly improves genetic resources available for tomato, but also provides new tools to study gene evolution and gene duplications as well as to identify paralogues. It is important to use other means than only sequence similarity in functional annotation of genes, for example gene expression patterns as in this manuscript. In addition to the newly produced sequence data, organization and classification of existing data greatly improves the usability of data from earlier studies.

The manuscript is well written and clear, with only minor grammatical errors in the text. The analyses are sound, results are clearly presented and methods explained in sufficient detail. I therefore only have minor suggestions to improve the manuscript. The most important point is to make clear the amount of newly sequenced data. At the moment it is not clear what data was downloaded from GenBank and what was produced for the current study.  

Material and methods, section 3.1 sample collection: The number of total samples sequenced for this study should be clearly mentioned.

Material and methods, sections 3.2 and 3.3 could be merged.

Material and methods, section 3.3: The number of libraries and sequenced lanes should be mentioned.

Line:257 something is missing after “possible”. Possible artifacts?

Material and methods, sections 3.6.-3.8: These sections are all on gene expression and could be merged. Section 3.10 could also be merged into the same section or at least be presented before 3.9. Please explain the shortening TGR when first mentioned.

L: 277: change licopersicum to lycopersicum and write in italics.

Data availability: The project PRJNA507622 is not found in the SRA database, please check it.

Author Response

The manuscript “Comparative analysis of development transcriptome maps of Arabidopsis thaliana and “Solanum lycopersicum” by Penin et al. constructs a transcriptomic expression database for tomato. The results of the manuscript greatly improves genetic resources available for tomato, but also provides new tools to study gene evolution and gene duplications as well as to identify paralogues. It is important to use other means than only sequence similarity in functional annotation of genes, for example gene expression patterns as in this manuscript. In addition to the newly produced sequence data, organization and classification of existing data greatly improves the usability of data from earlier studies.

The manuscript is well written and clear, with only minor grammatical errors in the text. The analyses are sound, results are clearly presented and methods explained in sufficient detail. I therefore only have minor suggestions to improve the manuscript. The most important point is to make clear the amount of newly sequenced data. At the moment it is not clear what data was downloaded from GenBank and what was produced for the current study. 

The amount of newly sequenced data is 60 libraries (30 samples in 2 replicates each). The samples are listed in the Supplementary Table 1.

Material and methods, section 3.1 sample collection: The number of total samples sequenced for this study should be clearly mentioned.

Material and methods, sections 3.2 and 3.3 could be merged.

Material and methods, section 3.3: The number of libraries and sequenced lanes should be mentioned.

This is done.

Line:257 something is missing after “possible”. Possible artifacts?

Indeed, we missed the word “artefacts”. This is corrected now.

Material and methods, sections 3.6.-3.8: These sections are all on gene expression and could be merged.

This is done.

Section 3.10 could also be merged into the same section or at least be presented before 3.9.

Please explain the shortening TGR when first mentioned.

This is done.

L: 277: Change licopersicum to lycopersicum and write in italics.

This is done.

Data availability: The project PRJNA507622 is not found in the SRA database, please check it.

It was closed, currently we released it: https://www.ncbi.nlm.nih.gov/bioproject/PRJNA507622

Round  2

Reviewer 1 Report

The manuscript has been significantly improved and now warrants publication in Genes